# Characterization of the Mechanism of Action of *Serratia rubidaea* Mar61-01 against *Botrytis cinerea* in Strawberries

**DOI:** 10.3390/plants12010154

**Published:** 2022-12-29

**Authors:** Zahra Alijani, Jahanshir Amini, Kaivan Karimi, Ilaria Pertot

**Affiliations:** 1Department of Plant Protection, Faculty of Agriculture, University of Kurdistan, Sanandaj P.O. Box 416, Iran; 2Safiabad Agricultural Research and Education and Natural Resources Center, Agricultural Research, Education and Extension Organization (AREEO), Dezful P.O. Box 333, Iran; 3Research and Innovation Center, Fondazione Edmund Mach (FEM), 38010 San Michele all’Adige, Italy; 4Center Agriculture Food Environment (C3A), University of Trento, 38010 San Michele all’Adige, Italy

**Keywords:** antioxidant enzymes, gray mold of strawberries, prodigiosin, *Serratia rubidaea*

## Abstract

Several bacterial strains belonging to *Serratia* spp. possess biocontrol capability, both against phytopathogens and human pathogenic species, thanks to the production of secondary metabolites, including as a red-pink, non-diffusible pigment, 2-methyl-3-pentyl-6-methoxyprodiginine (prodigiosin). *Botrytis cinerea* is the causal agent of gray mold, which is an economically relevant disease of many crops worldwide. Gray mold is normally controlled by chemical fungicides, but the environmental and health concerns about the overuse of pesticides call for environmentally friendly approaches, such as the use of biocontrol agents. In this study, the efficacy of a specific strain of *Serratia rubidaea* (Mar61-01) and its metabolite prodigiosin were assessed against *B. cinerea* under in vitro and in vivo conditions. This strain was effective against *B. cinerea*, and the effect of prodigiosin was confirmed under in vitro and in vivo conditions. The strain suppressed mycelial growth of *B. cinerea* (71.72%) in the dual-culture method. The volatile compounds produced by the strain inhibited mycelial growth and conidia germination of *B. cinerea* by 65.01% and 71.63%, respectively. Efficacy of prodigiosin produced by *S. rubidaea* Mar61-01 on mycelial biomass of *B. cinerea* was 94.15% at the highest concentration tested (420 µg/mL). The effect of prodigiosin on plant enzymes associated with induction of resistance was also studied, indicating that the activity of polyphenol oxidase (PPO), superoxide dismutase (SOD) and phenylalanine ammonia lyase (PAL) were increased when prodigiosin was added to the *B. cinerea* inoculum on strawberry fruits, while catalase (CAT) and peroxidase (POD) did not change. In addition, the volatile organic compounds (VOCs) produced by *S. rubidaea* Mar61-01 reduced mycelial growth and inhibited conidial germination of *B. cinerea* in vitro. The findings confirmed the relevant role of prodigiosin produced by *S. rubidaea* Mar61-01 in the biocontrol of *B. cinerea* of strawberries, but also indicate that there are multiple mechanisms of action, where the VOCs produced by the bacterium and the plant-defense reaction may contribute to the control of the phytopathogen. *Serratia rubidaea* Mar61-01 could be a suitable strain, both to enlarge our knowledge about the potential of *Serratia* as a biocontrol agent of *B. cinerea* and to develop new biofungicides to protect strawberries in post-harvest biocontrol.

## 1. Introduction

*Serratia* species are gram-negative bacteria belonging to Enterobacteriaceae and are commonly found in the environment (i.e., water and soil), plants, and insects. They can also infect vertebrates, including humans [1]. Some strains in the genus *Serratia* display potential biocontrol activities, both against important phytopathogens of main crops and species pathogenic to humans [2]. *Serratia* strains produce secondary metabolites that include a wide range of natural bioactive products, such as the red-pink, non-diffusible pigment, 2-methyl-3-pentyl-6-methoxyprodiginine, called prodigiosin [2]. Secretion of prodigiosin is a frequent trait of the genus *Serratia*. Prodigiosin is produced by four species, namely *S. marcescens*, *S. plymuthica*, *S. nematodiphila* and *S. rubidaea* [1]. Prodigiosin displays antibacterial and antifungal activity against several plant pathogens [1]. For example, antifungal effects against *Fusarium oxysporum*, *Rhizoctonia solani*, *Phytophthora parasitica* and *Mycosphaerella fijiensis* are associated with the production of prodigiosin by *S. marcescens* [3,4,5]. Prodigiosin prevents the mycelial growth and conidial germination of *Colletotrichum gloeosporioides* [6]. 

*Botrytis cinerea* is a necrotrophic pathogen of strawberry fruits, responsible for important economic losses in agriculture [7,8]. Several strategies have been used to control this pathogen, including cultural and chemical methods. *Botrytis cinerea* has high genetic variability, abundant reproduction and a short life cycle, which make it a high-risk pathogen, especially in terms of developing resistance to fungicides [9]. On the other hand, excessive use of fungicides on vegetables and fruits poses serious problems for the environment and human health [10]. Among biocontrol agents, endophytic bacteria have remarkable potential in the management of plant diseases [11] and numerous studies have reported using them for the successful control of plant pathogens [12,13,14,15]. Endophytic bacteria inhabit plant internal tissues without causing any noxious effects for their hosts [16,17], but can also exert positive effects for the plant: some endophytic bacteria can increase plant growth by promoting of nutrient uptake and phytohormone production [18,19], while others can enhance the plant immune system and suppress pathogens through the production of antifungal metabolites (antibiotics and lytic enzymes) and volatile organic compounds (VOC) [20,21,22].

Several strains belonging to *S. rubidaea* can protect plants against phytopathogens: for example, *S. rubidaea* S55 reduced foot and root rot of tomato caused by *F. oxysporum* f. sp. *radices-lycopersici* under in vitro and greenhouse conditions [23], and *S. rubidaea* Mar61-01 and prodigiosin extracted from this strain display antifungal properties against *C. nymphaeae* on strawberry [24].

One of the limiting factors in developing *Serratia* spp. candidates for biocontrol strategies against fungal pathogens is the scarce availability of studies, mainly due to the lack of characterized isolates [2], therefore clarifying the mechanism of action of new strains with biocontrol properties can positively contribute to the sector of microbial biofungicides. *Serratia rubidaea* Mar61-01 was isolated as an endophyte from healthy strawberry stems. The main mechanism for its ability to reduce infection of strawberry plant tissues by *C. nymphaeae,* both in vitro and in vivo, is due to the antibiosis exerted by prodigiosin [24]. Although the efficacy of this pigment of *S. rubidaea* against *C. nymphaeae* was demonstrated, the ability to enlarge its spectrum of activity against the main disease culprit in strawberries (*B. cinerea*) would make this strain a more appealing candidate for use as a biological control agent in strawberry production. In addition, further work is required to elucidate the interaction of Mar61-01 with the plant, including its effects on the plant defense system.

In the plant–pathogen–biocontrol agent interaction, plant enzymes (i.e., catalase, peroxidase, superoxide dismutase, phenylalanine ammonia lyase) may change, thus resulting in increased resistance against the pathogen as reported in previous studies, for example in the interactions among cucumber–*Cucumber mosaic virus*–*Trichoderma asperellum*, rice-maize–*R. solani*–rhizobacteria or cotton–*R. solani*–*T. virens* [25,26,27]. In this context, understanding if prodigiosin extracted from *S. rubidaea* Mar61-0 can modify the production of plant enzymes associated with resistance induction in strawberry fruits could help to elucidate the mechanisms involved in the three-way cross talk between the fruit, the pathogen and the antagonist. Therefore, the final goal of this research is to characterize the mechanism of action of *S. rubidaea* Mar61-01 to suppress the growth of *B. cinerea* under in vitro and in vivo conditions, and to study the effects of prodigiosin extracted from this bacterial strain on the enzymatic activity of strawberry fruits in interaction with the pathogen.

## 2. Results

### 2.1. Antifungal Activity of Serratia rubidaea Mar61-01 against Mycelial Growth of Botrytis cinerea

According to in vitro tests, the bacterial strain *S. rubidaea* MarR61-01 possesses antifungal effects against *B. cinerea*. T-test analysis showed a significant inhibitory effect of *S. rubidaea* MarR61-01 against the growth of the mycelium of *B. cinerea* in dual-culture tests (t (6) = −8.828, *p* < 0.0001); this was also confirmed by the paper-disc method with a significant reduction of the colony diameter (t (4) = 36.631, *p* < 0.0001), corresponding to an inhibition of 71.72% compared with control (Table 1). Observations under the microscope clearly confirmed these measurements; the length of the hyphae appeared reduced and thinner in comparison with the control (Appendix A). Curling, twisting (Appendix A) and lysis (data not shown) of fungal mycelia were also observed in the treated *B. cinerea*.

### 2.2. Effects of Volatile Organic Compounds of Serratia rubidaea Mar61-01 on Botrytis cinerea Mycelial Growth and Conidia Germination

The inhibition by *S. rubidaea* MarR61-01 is not only associated with metabolites diffused in the agar medium. VOCs of *S. rubidaea* MarR61-01 also play a significant role in reducing both the mycelial growth (t (4) = 8.345, *p* < 0.001) and the conidia germination (t (4) = 5.365, *p* < 0.006) of *B. cinerea* (Table 2). In fact, VOCs of *S. rubidaea* MarR61-01 showed inhibitory effects (71.63%) on conidial germination of *B. cinerea* on water agar medium (Table 2). Furthermore, mycelial growth of *B. cinerea* was also affected (Table 2).

### 2.3. Effects of Prodigiosin on Mycelial Growth and Biomass of Botrytis cinerea

Data showed that prodigiosin inhibited the mycelial growth of *B. cinerea* around the holes in Petri plates (Appendix A). Prodigiosin at the various tested concentrations decreased the biomass of *B. cinerea* compared to control in the liquid medium (Table 3). The inhibition is dose-dependent, and the greatest effect was noticed at concentrations from, or above, 220 µg/mL (Table 3). 

### 2.4. Protection of Strawberry Fruits against Botrytis cinerea by Serratia rubidaea MarR61-01 Cells

The efficacy of live cells of *S. rubidaea* MarR61-01 to control gray mold was estimated on strawberry fruits artificially inoculated with *B. cinerea*. In fruits treated by the bacterial cell suspension, the average disease severity of the treatment five days after pathogen inoculation was 0.1 and significantly (t (16) = 7.305, *p* < 0.0001) less than of positive control (0.28). The *S. rubidaea* MarR61-01 cells decreased disease severity by 64.28% compared to control, similar to the VOCs that significantly (t (10) = 3.945, *p* < 0.003) reduced the disease severity and displayed a biocontrol efficacy of 63.33% at a similar level of disease severity reduction (Table 4).

### 2.5. Effects of Prodigiosin Pigment on Activities of Defense-Related Enzymes

The activities of CAT, POD, PPO, SOD and PAL enzymes were evaluated in treated strawberry fruits to understand if prodigiosin produced by *S. rubidaea* MarR61-01 can induce resistance against *B. cinerea* in comparison to untreated fruits. The results indicate that *S. rubidaea* has a different effect on the activity of these enzymes (Figure 1).

Immediately after fruit inoculation with *B. cinerea* (0 h), CAT activity significantly increased compared to the control, while the treatments with prodigiosin and the combination of prodigiosin +*B. cinerea* did not. After 24 h, this increase was no longer present and all treatments did not show differences with the control until 96 h, when a limited, although significant, increase was noticed on fruits that received *B. cinerea* alone, or in combination with prodigiosin (Figure 1a).

The activity of POD in *B. cinerea* significantly increased immediately after fruit inoculation (0 h) compared to the control. This increase was observed again in the next 24 h and then decreased. In addition, after 72 h the treatments with prodigiosin increased and then decreased (Figure 1b).

The pattern of PPO activity was similar to the one of CAT at T0: PPO activity strongly increased in the fruits inoculated with *B. cinerea* (P), but not when prodigiosin was added to the pathogen (BP). In *B. cinerea* (P) treatment, POD decreased after 24 h (T24) then slightly, but significantly, increased from T72. On the other hand, PPO activity strongly increased after 96 h in +prodigiosin +*B. cinerea* (BP). PPO activation by *B. cinerea* was counteracted by the addition of prodigiosin in the short term; however, this was insufficient in the long term (T96). It seems that the effect of prodigiosin when added to *B. cinerea* is to delay the PPO increase due to *B. cinerea* (Figure 1c).

The antioxidative activity of SOD increased after 72 h, but only on fruits treated with prodigiosin when inoculated with (BP) or without *B. cinerea* (B) (Figure 1d).

The PAL activity started to increase after 24 h on fruits treated with +prodigiosin +*B. cinerea* (BP) and +prodigiosin (B) with a maximum at 96 h. The inoculation with *B. cinerea* did not increase PAL at any sampling time, which resulted in even lower levels than the control after 72 h (Figure 1e). 

The disease severity in prodigiosin-treated fruits was lower compared to control (+ *B. cinerea*) (Appendix A).

At the first sampling time, total phenol content did not differ among treatments. The total phenol content of the control and when fruits were treated with +prodigiosin (B) or inoculated with +*B. cinerea* (P) increased after 24 h, but not when prodigiosin and *B. cinerea* were combined (BP); they then decreased at 72 h to even lower levels than the combination (BP). At the last sampling time, only the fruits inoculated with *B. cinerea* displayed a lower content of total phenols compared to the treatment (Figure 1f).

## 3. Discussion

Based on the results of this research, *S. rubidaea* Mar61-01 can be considered a promising biocontrol agent against *B. cinerea* and could represent a new tool to control strawberry gray mold. Previous studies reported that another prodigiosin-producing strain, *S. rubidaea* S55, has antifungal effects against the plant pathogen *F. oxysporum* f. sp. *radicis-lycopersici* [23]. *Serratia rubidaea* MarR61-01 was isolated as a plant endophyte and displayed remarkable biocontrol activity against *C. nymphaeae* on strawberry plants, where it can efficiently control anthracnose in vitro, in vivo and under greenhouse conditions [24].

Biocontrol bacteria often possess complex mechanisms, and an accurate understanding of how they function can lead to their successful application as biological control agents [28]. The mechanism of action of *S. rubidaea* MarR61-01 against phytopathogens is already partially characterized: it is able to produce prodigiosin, protease and siderophores. It forms biofilm, produces IAA and GA, and is capable of phosphate solubilization and nitrogen fixation that can promote plant growth [8]. However, the mechanism involved in controlling postharvest diseases of strawberry fruits by endophytic bacteria can be quite complex, and previous studies suggest that production of VOCs [29] and disease resistance on the plant may be important components of the biocontrol [30]. Therefore, after verifying the antifungal activity of *S. rubidaea* MarR61-01 and prodigiosin against a worldwide important pathogen of strawberry (*B. cinerea*), we focused our study on the antifungal properties of VOCs emitted by *S. rubidaea* MarR61-01 and on understanding the impact of prodigiosin on some defense-related enzymes in the fruit.

Our in vitro studies (dual-culture and paper test) demonstrated that besides *C. nymphaeae* [24], *S. rubidaea* Mar61-01 can also control *B. cinerea*, thus enlarging the spectrum of activity of the bacterium and increasing the attractiveness of this strain as a biofungicide. Similar to the strawberry/*C. nymphaeae* pathosystem, prodigiosin confirmed its important role also in the antibiosis against *B. cinerea*. Nevertheless, *B. cinerea* seems to be more susceptible to the antifungal effect of prodigiosin than *C. nymphaeae*; in fact prodigiosin already shows a high inhibitory efficacy at 120 µg/mL, while a similar level of inhibition against *C. nymphaeae* is noticed only with concentrations almost ten times higher [24].

The antifungal properties of crude and purified prodigiosin were estimated against *B. cinerea* in jellified and liquid medium. In Petri plates on the jellified medium, inhibition zones around holes were observed and the addition of prodigiosin in the liquid medium reduced the biomass of *B. cinerea*. This is in agreement with Dawoud et al. [31] who demonstrated that crude and purified prodigiosin produced by *Bacillus* sp. DBS4 have antifungal activities against three fungi: *R. solani*, *F. oxysporum*, and *Sclerotium rolfsii*. In this study, the length of the hyphae was decreased, as compared to that in the control, along with shrinkage of hyphae. Curling and lysis of fungal mycelia were also noted. In addition, previous investigation showed that *S. marcescens* reduced infection of *P. capsici* on cucumber and these antifungal effects were not seen in pigment-defective mutants [3]. The production of prodigiosin by *S. rubidaea* Mar61-01 is therefore aconfirmed to also have a key role in the antifungal effect of this strain against *B. cinerea* [24].

In the sealed-plate test, VOCs produced by *S. rubidaea* MarR61-01 significantly reduced mycelial growth and conidia germination of *B. cinerea*, indicating an important antifungal activity of these compounds. Previous research has shown that the VOC produced by *Serratia* spp. is sodorifen. The VOCs can act as signals for communication between different organisms or serve as attractant or defense compounds [2]. It has been shown that, *Serratia plymuthica* PRI-2C produced sodorifen when exposed to the phytopathogen *F. culmorum* [32].

Generally, some of the compounds increasing fruit resistance to pathogens have an antioxidant nature [33], and various other biological elicitors, such as, chitosan, melatonin, BTH, etc. are known [34,35,36]. For example, a *Morchella conica* mycelial extract led to upregulation of genes related to defense against biotic stress in the treated fruit [37]. It is already known that plants that are infected with pathogens after exposure to biological control agents may present high levels of catalase, peroxidase, polyphenol oxidases, superoxide dismutase and phenylalanine ammonia-lyase enzymes [38]. In the present research, we found that the red pigment produced by *S. rubidaea* MarR61-01 changes the activity of some defense-related enzyme in the treated fruit.

CAT converts the unstable and toxic reactive oxygen species (ROS) to less toxic and stable components. CAT is an important and highly active enzyme in organisms. Several studies have presented that overexpression of CAT can increase plant resistance to abiotic and biotic stresses and is an important enzyme for reducing oxidative stress [39]. CAT activity was enhanced in fruits treated with +prodigiosin (B) at all time-points, and with +prodigiosin+*B. cinerea* (BP) treatment 0, 24 and 96 h post-inoculation compared to control. It was demonstrated that this enzyme increases cell wall resistance and induces defensive gene expression as a signal [40].

POD is a dual-function enzyme that oxidizes different substrates in the presence of H_2_O_2_ and produces reactive oxygen species. POD converts some carbohydrates into lignin, in this way enhancing the degree of lignification in fruit. Hence, high POD activity is associated with the beginning of induced resistance. In addition, POD eliminates toxic H_2_O_2_, phenols, amines, aldehydes and benzene [41]. After three days, POD activity was enhanced in the +prodigiosin (B)-treated group. The POD activity then remained low at the end of the assay which is consistent with the study by Ye et al. [41].

The PPO enzyme catalyzes the oxidation of phenolics to quinines, which are highly toxic to fungal pathogens. Enhanced PPO enzyme activity is associated with resistance of disease in fruit tissues. PPO enzyme activity of the +prodigiosin+*B. cinerea* (BP)-treated fruits increased at four days after inoculation, which was higher than the control., A previous study also showed that the activity of the PPO enzyme was enhanced in peaches by *Pichia membranaefaciens* to contribute to the defense against *Rhizopus stolonifer* [42].

SOD is an important enzyme in the defense pathways, that dismutases superoxides into O_2_ and H_2_O_2_. SOD has been proposed to be essential in plant stress resistance and creates the first line of defense against the adverse effects of high levels of ROS [39]. In our research, SOD activity increased in fruits inoculated with the +prodigiosin (B) at the 96 h time-point, and in fruits inoculated with the +prodigiosin+*B. cinerea* (BP) at 72 and 96 h post inoculation. Rais et al. [43] showed that antagonistic bacteria induced SOD activity in rice.

PAL is the main enzyme that catalyzes the metabolic reaction of phenylpropanoid and the phenylpropanoid metabolic pathway is activated in plants infected with pathogens [41]. The change in PAL enzyme activity is related to the degree of plant resistance along with total phenol content. PAL plays main roles in biotic and abiotic stress responses in plants and there are several studies that show that PAL genes are involved in the response of plants to infection by pathogens. PAL catalyzes the first step of the phenylpropanoid pathway and the synthesis of diverse natural products based on the phenylpropane skeleton, such as hydroxycinnamic acid and flavonoids. Phenylpropanoid compounds have been shown to play important roles in plant defense to phytopathogens, based on the correlation between rates of phenylpropanoid accumulation and expression of resistance in vivo [25,41]. The +prodigiosin (B) and +prodigiosin*+B. cinerea* (BP)-treated groups enhanced PAL enzyme activity after 2 d, which was higher than that of the control and demonstrated that +prodigiosin (B) and +prodigiosin*+B. cinerea* (BP) treatments could induce resistance in strawberry fruits.

Phenolic acids contribute significantly to the total antioxidant activity. Phenolic compounds induced in host plants are directly toxic or mediate the signaling of several transduction pathways, which produce toxic secondary metabolites and activate defense enzymes [39]. The high levels of total phenol content in the +prodigiosin (B) and +prodigiosin+*B. cinerea* (BP)-treated strawberry fruits in comparison with +*B. cinerea* (P)-treated group may be related to the fact that higher levels of phenolic compounds can act as signaling molecules to inhibit fruit decay. In agreement with our findings, changes in total phenolic content were shown when an *M. conica* mycelial extract was applied to the strawberry fruit surface [37].

These results suggested that the activities of these five defense-related enzymes (CAT, POD, PPO, SOD and PAL) in strawberry fruits were all induced by prodigiosin produced by *S. rubidaea*. In addition, the disease severity in prodigiosin-treated fruits was lower compared to control (+*B. cinerea*). In summary, the results showed that the activities of CAT, PPO, SOD and PAL were influenced by the treatments. This indicates that prodigiosin may directly counteract *B. cinerea* by antibiosis, but also induce a defense response in the fruit, suggesting a complex mechanism of action, in agreement with previous studies carried in a different pathosystem [21]. Indeed, in addition to the toxic effect of prodigiosin and VOCs, *S. rubidaea* MarR61-01 is capable of secreting protease enzymes in vitro [24], and lytic enzymes with a central role in the degradation of fungal cell wall [44] or the proteinaceous content of fungal cell walls and cytoplasmic proteins [45]. *Serratia rubidaea* MarR61-01 is capable of producing biofilm and protease enzymes that help the colonization of fruit tissues and, similar to another strain (*S. marcescens* AL2-16), also produces siderophores and phytohormones [24,46]. Siderophores are small proteins that have an efficient role in host plant defense against fungal pathogens [47] and phytohormones are important signaling molecules produced by biological control agents which are involved in regulating plant growth. For example, IAA has positive effects on root morphology and growth of plants [48]. Gibberellins (GAs) also increase the root area and number of root tips [49]. Several research studies have shown that some bacteria producing siderophores and phytohormones promote plant growth and rooting, including *B. subtilis* and *Stenotrophomona maltophilia* [50]. Disease severity of gray mold under in vivo experiments was reduced significantly by the living cells or prodigiosin and VOCs produced by bacterium; however, we cannot exclude the elicitation of defense mechanisms in fruits by digestion of the cell wall and secret endogenous elicitors [51], in addition to the contribution of other components such as siderophores, phytohormones and enzymes.

## 4. Materials and Methods

### 4.1. Bacterial and Fungal Strains and Plant Material

A sample of the sendophytic bacterium *S. rubidaea* Mar61-01 (GenBank accession no. MK880635), originally isolated from the stem of *Fragaria* × *ananassa* cv. Paros [24], was obtained from the culture collection of the University of Kurdistan in Iran and stored long-term in nutrient broth (NB; Merck, Darmstadt, Germany) supplemented with 20% (*v*/*v*) glycerol (Merck, Germany) at −20 °C. In the experiments, fresh bacterial cultures were used and obtained by incubating bacterial cells in Nutrient Agar (NA; Merck, Germany) for 24 h at 28 ± 2 °C.

*Botrytis cinerea* was acquired from the same culture collection and cultured on potato dextrose agar (PDA; Merck, Darmstadt, Germany) for 4 days at 25 ± 2 °C. Mature and healthy strawberry fruits (cv. Paros) having similar size (approximately 30 g) and age (similar color) were obtained from untreated and symptomless plants maintained in the greenhouse. In the biocontrol experiments the detached fruits were incubated in a growth chamber under a 16/8 h light/dark cycle at 25 ± 2 °C under continuous illumination with cool white fluorescent tubes at an intensity of 2000 lux.

### 4.2. Antifungal Activity of Serratia rubidaea Mar61-01 against Mycelial Growth of Botrytis cinerea

Two types of experiments were carried out: dual-culture test and paper-disc test. For the dual-culture test, *S. rubidaea* Mar61-01 was streaked on one side of a Petri plate (90 mm) containing PDA and a 5-mm diameter disc of a 4-day-old *B. cinerea* culture was placed at one side of the plate at approximately 2 cm from the *S. rubidaea* Mar61-01 culture line. The control was represented by PDA inoculated only with fungal pathogen [52]. The Petri plates were then incubated at 25 ± 2 °C for 5 days in the dark and the distance between the fungal colony and bacterial culture (inhibition zone) was measured. Four replicates (Petri plates) were used for each treatment.

For the paper-disc test, *S. rubidaea* Mar61-01 was incubated in LB broth (Merck, Darmstadt, Germany) at 28 ± 2 °C for 72 h and the final concentration of culture was adjusted to (1 × 10^8^ CFU/mL) with sterile LB. A 5-mm diameter disc of *B. cinerea* was obtained from a 4-day-culture and placed on the surface of a Petri plate containing PDA. Five-mm diameter discs of sterile filter paper (Merck, Germany) were impregnated individually with 20 μL of the *S. rubidaea* MarR61-01 culture broth with cells and placed at a distance of 2 cm from the *B. cinerea* culture disc. The plates were incubated at 25 ± 2 °C for 4 days in the dark and checked daily. Sterile filter-paper discs were impregnated with sterile 20 μL LB broth and used as controls [41]. Four replicates (Petri plates) were used for each treatment. The mycelial growth of fungal pathogen was assessed by AutoCAD software (San Francisco, CA, USA, version 2018) and the biocontrol efficiency was calculated [35] as percentage (I_m_) using the following formula:I_m_ (%) = [(A_c_ − A_t_)/A_c_] × 100
where A_c_ is area of colony of pathogen in control, A_t_ is area of colony of pathogen in treatment.

### 4.3. Effects of Volatile Organic Compounds of Serratia rubidaea Mar61-01 on Botrytis cinerea Mycelial Growth and Conidia Germination

To assess the effects of VOCs emitted by *S. rubidaea* MarR61-01, the double Petri plate system was used [53]. Briefly, the bacterial strain was cultured for 24 h on NA and a 5-mm disc of a 4-day-old culture of *B. cinerea* was placed at the center of a Petri plate containing PDA. Afterwards, the plate containing the *B. cinerea* disc was placed inversely on the plate containing *S. rubidaea* MarR61-01 and sealed with Parafilm. PDA plates containing only *B. cinerea* and coupled with Petri plates containing sterile NA were used as controls. The coupled Petri plates were incubated at 25 ± 2 °C for 5 days in the dark. Then, the growth of mycelial of fungal pathogen was assessed and the biocontrol efficiency was calculated as reported above. 

The effects of VOCs emitted by *S. rubidaea* MarR61-01 on conidial germination of *B. cinerea* were assessed by streaking 50 µL of a conidial suspension of *B. cinerea* (1 × 10^6^ CFU/mL) obtained from conidiating colonies on water agar (WA; Merck, Darmstadt, Germany). In addition, 150 µL of *S. rubidaea* MarR61-01 cells suspension (1 × 10^8^ CFU/mL) was streaked on NA. The Petri plates containing *B. cinerea* were placed above the plates containing *S. rubidaea* MarR61-01 and sealed using Parafilm. Petri plates containing *B. cinerea* combined with plates without bacterial strain were used as controls. The plates were incubated at 25 ± 2 °C for 72 h in the dark [54]. The inhibition percentage of conidial germination (I_c_) was measured according to the following formula:I_c_ (%) = [(C_c_ − C_t_)/C_c_] × 100
where C_c_ is number of germinated conidia in control, C_t_ is number of germinated conidia in treatment. Three replications were considered for each treatment. In two tests, three replications were considered for each treatment

### 4.4. Extraction of Prodigiosin 

Prodigiosin secreted by *S. rubidaea* MarR61-01 was extracted and determined using the method described by [24]. Briefly, *S. rubidaea* MarR61-01 was inoculated in sterile NB and incubated at 28 ± 2 °C on a rotary shaker (PIT10LO, Pol Ideal Pars Co., Tehran, Iran) at 150 rpm for 72 h. The culture medium (100 mL) was centrifuged at 10,000× *g* for 20 min (MIKRO 200R, Hettich, Kirchlengern, Germany), then mixed with acidified methanol (Merck, Germany; Ph = 3) and centrifuged at 5000× *g* for 15 min. The resulting supernatant was concentrated in a rotary evaporator (Z334898, Merck, Darmstadt, Germany) at 60 °C and the pigment was purified by thin layer chromatography (TLC; Merck, Germany) by silica gel (mesh size 80–100; Merck, Germany) as the solid matrix in a pre-saturated TLC chamber with mobile phase (chloroform/methanol in the ratio of 9:1 *v*/*v*).

### 4.5. Effects of Prodigiosin on Mycelial Growth and Biomass of Botrytis cinerea

The extracted prodigiosin was dissolved in sterile distilled water (at a concentration of 200 ppm) and then filtered using a 0.22 µm Millipore filter. The plates (90-mm diameter) were filled with PDA and inoculated by a conidia suspension of *B. cinerea* (100 µL of spore suspension with 1 × 10^6^ CFU/mL concentration was mixed into 5 mL of PDA and poured on the medium in the plates). In each plate, three 5-mm-diameter holes were created. To each hole, 150 µL of filtered red pigment solution was loaded. In control plates, each hole was filled with sterile distilled water. Plates were placed at 25 ± 2 °C for 72 h in the dark [24,55]. The inhibition zone around of holes was observed. 

The effects of prodigiosin produced by *S. Rubidaea* MarR61-01 were assayed on the biomass of *B. cinerea* according to reported methods [56]. Briefly, the powder of prodigiosin obtained with the extraction reported above was dissolved in sterile distilled water at concentrations of 0, 20, 120, 220, 320, and 420 µg/mL and then filtered using a 0.22 µm Millipore filter. The different concentrations were added to 100 mL of potato dextrose broth (PDB; Merck, Darmstadt, Germany) in Erlenmeyer flask (BRAND, Wertheim am Main, Germany) and then each Erlenmeyer flask was inoculated with one 5-mm-diameter plug of mycelial of a 4-day-old culture of *B. cinerea* and incubated on a rotary shaker (150 rpm) at 25 ± 2 °C for 14 days in the dark. An Erlenmeyer flask with PDB without prodigiosin was used as a control. Afterwards, the biomass in each treatment was separated by filtration on filter paper (No 1. Whatman; Merck, Germany) and the constant weight was measured with weighing scale (HR200, AND, Toshima-ku, Tokyo, Japan). Three replications (flasks) were performed for each treatment. The inhibition percentage of biomass (I_b_) was measured according to the following formula: I_b_ (%) = [(B_c_ − B_t_)/B_c_] × 100
where B_c_ is the biomass in control, B_t_ is biomass in treatment. 

### 4.6. Protection of Strawberry Fruits against Botrytis cinerea by Serratia rubidaea MarR61-01 Cells

The effects of the application of *S. rubidaea* MarR61-01 living cells on fruit decay development under in vivo conditions were evaluated. The strawberry fruits were rinsed with tap water for 30 s and sterilized in 70% ethanol (Merck, Darmstadt, Germany) for 30 s and then washed three times with sterile distilled water. Disinfected fruits were then dipped in the *S. rubidaea* MarR61-01 cells suspension (1 × 10^8^ CFU/mL) for 5 min and then allowed to dry on sterile filter paper for 30 min. Next, fruits were kept in covered plastic boxes (4 × 5 cm) for 24 h and one 5-mm-diameter plug of mycelial of a 4-day-old culture of *B. cinerea* was placed on the surface of each fruit. The fruits inoculated with sterile distilled water and only a plug of the fungal pathogen was used as negative and positive controls, respectively [57]. Nine replications (fruits) were performed for each treatment. The fruits placed into covered plastic boxes were incubated at 25 ± 2 °C, 75–80% relative humidity (RH) and checked daily until symptom expression.

### 4.7. Protection of Strawberry Fruits against Botrytis cinerea by Volatile Organic Compounds Emitted by Serratia rubidaea MarR61-01

Fruit protection test by *S. rubidaea* MarR61-01 VOCs was performed in closed glass desiccators (24 × 18 cm, diameter × height; ~6 L in volume). For this purpose, *S. rubidaea* MarR61-01 was cultured onto NA in Petri plates (diameter = 14 cm) and incubated at 28 ± 2 °C for 24 h. The uncovered plates containing bacterial cultures were placed at the bottom of the desiccators (one plate per desiccator). Strawberry fruits disinfected as described above were then inoculated with one 5-mm-diameter plug of mycelial of a 4-day-old culture of *B. cinerea*. Afterward, the inoculated fruits were placed on the perforated ceramic clapboard above the uncovered plates with the bacterial culture. Fruits inoculated with sterile distilled water or *B cinerea* above uncovered plates with only NA were used as negative and positive controls, respectively. All desiccators were covered with Parafilm and kept at 25 ± 2 °C under a regime of 12:12 h (light: dark) for 5 d [54]. 

In both experiments, sunken necrotic lesions on strawberry fruits were assessed and measured after 5 days (symptom appearance on the positive control); each strawberry fruit was considered as a conical shape and the total infected area was calculated by considering the radius and height of the fruit using AutoCAD software (San Francisco, CA, USA, version 2018). Disease severity (DS) was estimated by using the following formula [58]:DS = A/H × 2πr
where (A) is the infected area, (H) and (r) are height and the base radius of the fruit, respectively.

### 4.8. Effects of Prodigiosin on Activities of Plant Defense-Related Enzymes

The fresh strawberry fruits were hand-picked from the greenhouse and transferred to the laboratory within 30 min to be used in the experiments. Healthy strawberries were selected and collected based on being 70–80% ripened, and with uniform size (30 g) and shape. Strawberry fruits were surface-disinfected as described above. Four treatments were considered including mock-treated control (C), +*B. cinerea* (P), +prodigiosin (B) and +prodigiosin+*B. cinerea* (BP) at different sampling times (0, 24, 72 and 96 h after *B. cinerea* inoculation). Fruits were wounded by a sterile needle to a 2-mm depth and 20 µL filtered purified prodigiosin (at concentration of 200 ppm) was injected on the wound of each fruit in the treatment group; 20 µL sterile distilled water was used as control. Fruits were kept in covered plastic boxes (4 × 5 cm) at 25 ± 2 °C under 70–80% RH for 24 h. Afterward, one 5-mm-diameter plug of mycelial of a 4-day-old culture of *B. cinerea* was placed on each wound of each fruit, which were kept under the same conditions [37]. After the mentioned times, the samples were immediately frozen in liquid nitrogen and stored at −70 °C.

A small piece (1 g) of strawberry fruit tissue was taken from around the wound (8 mm diameter and 5 mm deep) of the various treatments at the various sampling times and homogenized in 5 mL of precooled phosphate buffer (4 °C, 100 mmol/L, pH 6.4) containing 0.5% polyvinyl pyrrolidone (Merck, Germany). After that, the ground tissues were centrifuged at 12,000× *g* for 20 min at 4 °C (MIKRO 200R, Hettich, Germany) and the supernatant (extract) used for determining the activities of defense-related enzymes namely, catalase (CAT), peroxidase (POD), polyphenol oxidase (PPO), superoxide dismutase (SOD) and phenylalanine ammonia lyase (PAL) enzymes. For CAT extraction, the tissues were macerated in precooled phosphate buffer (4 °C, 50 mmol/L, pH 7.0) [59]. All enzyme activity data represent the average of three independent samples in each period of time. The total soluble protein content was estimated for each treatment according to the Bradford assay [60] with bovine serum albumin (Sigma) as standard. All absorbances in this research were measured by Single Beam UV–Visible Spectrophotometer (SPECORD^®^ PC/210, Analytik Jena, Germany). To estimate CAT activity, the extract (500 μL) was mixed with 2 mL of phosphate buffer (50 mM, pH7.0) and then 500 μL of H_2_O_2_ (40 mM) added. The activity of CAT was determined by monitoring the absorbance at 240 nm (A240) with a spectrophotometer (SPECORD^®^ PC/210, Analytik Jena, Jena, Germany). The decomposition of H_2_O_2_ was evaluated as the change of A240 every 10 s for 3 min. The specific enzyme activity was expressed in units/mg protein, where one unit of catalase converts one μmol of H_2_O_2_/min [61].

To determine POD activity, the extract (500 µL) was poured into 2 mL of phosphate buffer (100 mM, pH 6.4) containing guaiacol (8 mM) and kept at 30 °C for 5 min. Afterward, 1 mL of H_2_O_2_ solution (24 mM) was added to the mixture. The POD activity was estimated by measuring the absorbance at 470 nm once every 30 s for 3 min, and the activity was expressed as ΔOD470/min/mg protein [62].

The activity of PPO was evaluated by measuring the change in absorbance at 495 nm for 3 min. Initially, the extract (200 µL) was added to 1.5 mL of sodium phosphate buffer (100 mM, pH 6.4) containing catechol (0.01 M) and incubated at 30 °C for 5 min. A495 was recorded once every 30 s for 3 min and the activity was expressed as ΔOD495/min/mg protein [42].

To determine SOD activity, the extract (100 µL) was added to 3 mL phosphate buffer (50 mM, pH 7.8) containing methionine (13 mM), nitro-blue tetrazolium (NBT) (75 μM), Ethylenediaminetetraacetic acid (EDTA) (10 μM) and riboflavin (2 μM). The mixture was incubated at 4000 lux for 10 min, and the absorbance was recorded at 560 nm. The specific activity was expressed as units/mg protein with one unit of SOD defined as the amount of enzyme that caused a 50% decrease in the SOD-inhabitable NBT reduction [63].

To determine PAL activity, strawberry fruit tissue (1 g) was macerated in acetone (10 mL) and left for 15 min at room temperature, then filtered by Whatman paper and the extract was dried in an oven (Cole Parmer, GZ-05012-41, Cambridgeshire, England) at 40 °C. The resulting powder (0.1 g) was mixed with 1 mL sodium borate buffer (0.1 M, pH 8.8) and kept at 4 °C for 45 min. The mixture was centrifuged at 10,000× *g* at 4 °C for 45 min and the supernatant was precipitated by ammonium sulfate (27%). Afterward, 4.5 mL ammonium acetate buffer (0.1 M, pH 7.7) was poured onto the pellet and mixed with 3.4 mL deionized water and 600 μL of L-phenylalanine solution (100 mM). Then, the mixture was incubated in a water bath at 40 °C for 45 min. The PAL activity was evaluated by measuring the absorbance at 290 nm and expressed as micromoles of cinnamic acid/mg of protein [38].

For the total phenolic assessment, 0.5 g strawberry fruits from treated and control groups were macerated in acidified methanol (HCl/methanol/water, 1:80:10, *v*/*v*), and the centrifuged at 10,000× *g* for 20 min. Then, five mL sterile distilled water and 250 μL Folin–Ciocalteu reagent were added to 1 mL supernatant. After mixing, extracts were kept for 5 min at room temperature and then 1 mL sodium carbonate (7.5%) and 1 mL sterile distilled water were added to each tube and incubated for 1 h at room temperature in the dark [64]. Then, absorbance was measured at 725 nm. Total phenolic content in the fruits were estimated as pyrogallol equivalents in mL/g frozen fruit.

### 4.9. Statistical Analysis

The completely randomized block design was used in all experiments except for the effects of prodigiosin on activities of plant defense-related enzymes section, in which a completely randomized factorial design with two treatments method was used. The experiments were repeated and performed with at least three replications. One representative experiment is presented in the results. After testing for homoscedasticity and normal distribution, the tests were designed as a completely randomized design and data were subjected to t-test and a one-way analysis of variance (ANOVA), and means separated according to LSD test using SAS software (version 8.2; SAS Institute, Cary, NC 27513, USA). All data are presented as mean values ± standard deviation (SD). Statistical significance between different treatments was assessed using LSD test (α = 0.05) [25,33].

## 5. Conclusions

In the present study, *S. rubidaea* Mar61-01 succeeded in controlling *B. cinerea*, the causal agent of strawberry gray mold. The results show not only that *S. rubidaea* Mar61-01 cells, VOCs, and the purified red pigment prodigiosin produced by it were able to decrease mycelial growth, conidial germination and fruit decay development under in vitro and in vivo experiments, but also that prodigiosin impacted the activity of some fruit enzymes associated with the defense mechanism and resistance to gray mold infection. Our findings demonstrate that prodigiosin affects the enzyme activity of strawberry fruits. However, further research is needed to understand the interaction of *S. rubidaea* Mar61-01 with the host and phytopathogen and its effects on the plant immune system.

## Figures and Tables

**Figure 1 plants-12-00154-f001:**
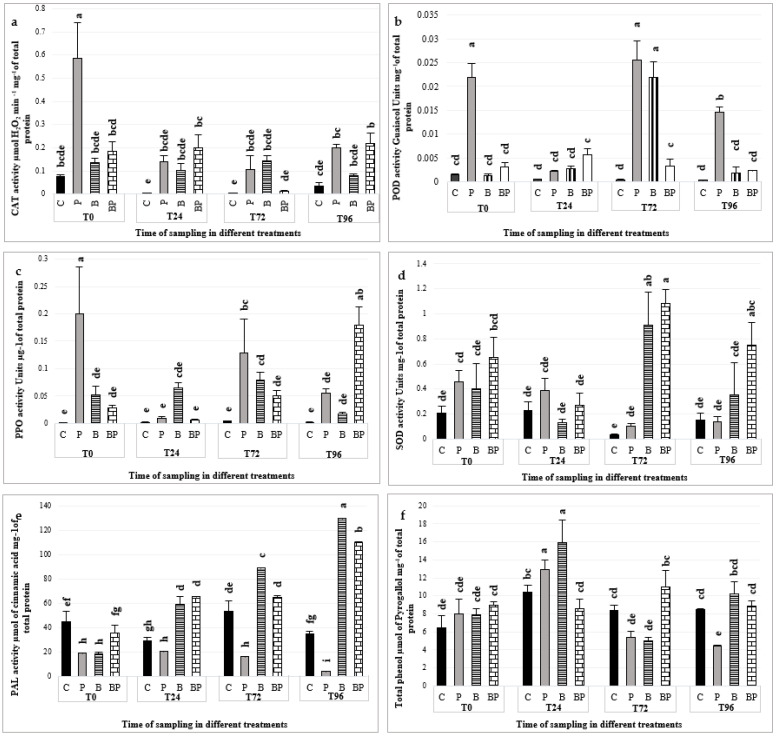
Activity of some defense-related enzymes in response to prodigiosin produced by *S. rubidaea* analyzed in mock-treated control fruits (C); + *B. cinerea* inoculated fruits (P); +prodigiosin inoculated on fruits (B); and +prodigiosin inoculated on fruits + *B. cinerea* treated plants (BP), at 0, 24, 72 and 96 h after *B. cinerea* infection. The bars are the mean activities of the defense-related enzymes CAT (**a**), POD (**b**), PPO (**c**), SOD (**d**), PAL (**e**) and total phenol (**f**) calculated on three replicates. Mean followed by different letters represents significant differences according to the LSD test (α = 0.05).

**Table 1 plants-12-00154-t001:** Inhibitory effect of *Serratia rubidaea* MarR61-01 against *Botrytis cinerea* under in vitro tests.

Treatments	Dual-Culture Test			Paper-Disc Test		
	Inhibition Zone (cm)	Inhibition I_m_ (%)	*t*-Value	Colony Diameter (cm^2^)	Inhibition I_m_ (%)	*t*-Value
MarR61-01	1.4 ± 0.29	-	−8.828 **	17.19 ± 1.50	71.72	36.63 **
Control	0.0 ± 0.00	-		60.80 ± 1.41	-	

Significant at *p* = 0.01 represented by **. I_m_: inhibition percentage of mycelial growth.

**Table 2 plants-12-00154-t002:** Inhibitory effect of VOCs produced by *Serratia rubidaea* MarR61-01 against *Botrytis cinerea* under in vitro tests.

Treatments	Mycelial Growth Test			Conidia Germination Test		
	Colony Diameter (cm^2^)	Inhibition I_m_ (%)	*t*-Value	Germinated Conidia (n)	Inhibition I_c_ (%)	*t*-Value
MarR61-01	17.14 ± 5.80	65.01	5.36	31 ± 12.00	71.63	8.34 **
Control	48.99 ± 8.40	-		109.3 ± 10.90	-	

Significant at *p* = 0.01 represented by **. I_m_: inhibition percentage of mycelial growth, I_c_: inhibition percentage of conidia germination.

**Table 3 plants-12-00154-t003:** Efficacy of prodigiosin produced by *Serratia rubidaea* Mar61-01 on mycelial biomass of *Botrytis cinerea*.

Concentration of Pigment (µg/mL)	Biomass Weight (gr)	Inhibition I_b_ (%)
0	3.08 ± 0.56 ^a^	-
20	0.62± 0.11 ^bc^	59.74
120	0.35 ± 0.037 ^cd^	88.63
220	0.19 ± 0.026 ^d^	93.83
320	0.196 ± 0.032 ^d^	93.63
420	0.18 ± 0.02 ^d^	94.15

Means followed by different letters within the column represent significant differences according to the LSD test (*α* = 0.05). Data are the mean of three replicates ± standard deviation (SD). I_b_: inhibition percentage of biomass.

**Table 4 plants-12-00154-t004:** Effects of *Serratia rubidaea* MarR61-01 on fruit decay caused by *Botrytis cinerea* assessed 5 days after treating fruits with the bacterial cells or volatile compounds produced by the bacterium. Biocontrol efficacy is expressed as reduction (%) of disease severity (% of fruit surface with gray mold symptoms) compared to untreated control.

Treatments	Living Cells			Volatile Compounds		
	Disease Severity	Biocontrol Efficacy (%)	*t*-Value	Disease Severity	Biocontrol Efficacy (%)	*t*-Value
MarR61-01	0.10 ± 0.04	64.28	7.3 **	0.11 ± 0.04	63.33	3.94 **
Control	0.28 ± 0.06	0		0.3 ± 0.11	-	

Significant at *p* = 0.01 represented by **.

## Data Availability

The raw data supporting the conclusion of the results will be made available on request.

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
