# Peer review of "Characterization of the Mechanism of Action of Serratia rubidaea Mar61-01 against Botrytis cinerea in Strawberries"

_plants, 2022, doi:10.3390/plants12010154_

Round 1
Reviewer 1 Report
Interesting work on the use of Serratia and its metabolite for the biocontrol of B. cinerea in strawberries. Here are some comments.
-Line 12: eliminate (J. Amini)
-Line 15: 2-methyl-3-pentyl-6- methoxyprodiginine > 2-methyl-3-pentyl-6-methoxyprodiginine
-Line 19-21: In this study, the efficacy against B. cinerea of a specific strain of Serratia rubidaea (Mar61-01) was assessed and the effect of prodigiosin against B. cinerea was confirmed under in vitro and in vivo conditions. > In this study, the efficacy of a specific strain of Serratia rubidaea (Mar61-01) and its metabolite prodigiosin were assessed against B. cinerea under in vitro and in vivo conditions.
-Line 30: Serratia rubidaea > S. rubidaea
-Line 33: could be added: B. cinerea, biofungicide, post-haverst biocontrol
-Line 42: same as line 15
-Line 52: sp. radices-lycopersici > sp. radices-lycopersici
-Line 70: Botrytis cinerea > B. cinerea
-Revise the names of the microorganisms, especially B. cinerea and Serratia spp. throughout the article.
-Include in the footnotes of the tables the meanings of Im, Ic,....
-Line 308: (v/v) in italics
-Line 398: How many replicates of this assay were performed? What parameters were measured? What variety of strawberry was used? What was its cultivation, characteristics, previous treatments? What was the size of the strawberries? All this information needs to be included.
Author Response
Plants
Manuscript ID: plants-2075738
Dear Reviewer
Hello and thanks for your help
We are grateful for your valuable comments and suggestions which helped us to improve the manuscript. We have taken care of all the corrections/ suggestions / comments and are ready to do any further changes that may be required. Following are our response to your queries:
Line 12: eliminate (J. Amini)
Response: j. amini was removed
Line 15: 2-methyl-3-pentyl-6- methoxyprodiginine
Response: this phrase was corrected
Line 19-21: In this study, the efficacy against B. cinerea of a specific strain of Serratia rubidaea (Mar61-01) was assessed and the effect of prodigiosin against B. cinerea was confirmed under in vitro and in vivo conditions. > In this study, the efficacy of a specific strain of Serratia rubidaea (Mar61-01) and its metabolite prodigiosin were assessed against B. cinerea under in vitro and in vivo conditions.
Response: this sentence was corrected within the text.
Line 30: Serratia rubidaea > S. rubidaea
Response: All names were abbreviated except at the beginning of the sentence. Actually we prefer is to adopt the standard editorial rule, where scientific names are not abbreviated at the beginning of sentences.
Line 33: could be added: B. cinerea, biofungicide, post-harvest biocontrol
Response: The words were added
Line 42: 2-methyl-3-pentyl-6- methoxyprodiginine
Response: this phrase was corrected
Line 52: Radices
Response: The mistake was corrected
Line 70: Botrytis cinerea > B. cinerea
Response: All names were abbreviated except at the beginning of the sentence
Tables: Include in the footnotes of the tables the meanings of Im, Ic,....
Response: Abbreviated terms explained
Line 308: v/v in italics
Response: v/v were corrected in text
-Line 398: How many replicates of this assay were performed? What parameters were measured? What variety of strawberry was used? What was its cultivation, characteristics, previous treatments? What was the size of the strawberries? All this information needs to be included.
Response: Information related to each experiment is mentioned separately in the materials and methods section.
We look forward to hearing from you regarding our submission. We would be glad to respond to any further questions and comments that you may have.
Sincerely,
Jahanshir Amini
Reviewer 2 Report
see attach file

Author Response
Plants
Manuscript ID: plants-2075738
Dear Reviewer
Hello and thanks for your help
We are grateful for your comments and suggestions which helped us to improve the manuscript indeed. We have taken care of all the corrections/ suggestions / comments and are ready to do any further changes that may be required. Following are our response to your queries:
Abstract: you should added some value of your results here
Response: Required values were added and highlighted in the text
Line 52: Radices-lycopersici
Response: The mistake was corrected
Line 59: Serratia rubidaea
Response: All names were abbreviated except at the beginning of the sentence. Actually we prefer is to adopt the standard editorial rule, where scientific names are not abbreviated at the beginning of sentences.
Line 68: I think this part should move before line 50
Response: This part was moved before line 50 and highlighted in the text
-Line 86: use other references for example bacteria against fungi
Response: we added two references as below:
- Bhattacharyya, C.; Banerjee, S.; Acharya, U.; Mitra, A.; Mallick, I.; Haldar, A.; Haldar, Sh.; Ghosh, A.; Ghosh, A. Evaluation of plant growth promotion properties and induction of antioxidative defense mechanism by tea rhizobacteria of Darjeeling, India. Sci Rep2020, 10(1), 1-19.
- Pratap, S.S.; Keswani, C.; Sansinenea, E.; Xuan, T.H. Trichoderma spp. mediated induction of systemic defense response in brinjal against Sclerotinia sclerotiorum. Curr Res Microbial Sci 2021, 2, 100051-100051.
Table 1: when you have two treatments you do not need to do LSD test you only can say this is significant or not
Response: Thank you for this constructive comment. T-test was done instead of LSD test.
Line 120: letters within the column represents significant differences according
Response: this sentence was corrected
Line 174: you can move to discussion part
Response: this paragraph was moved to discussion
Line 190: in general this discussion part need to improve it is very poor
Response: Added more details to this section
Line 270: what is the role of PAL you should discuss this
Response: the role of PAL was discussed in the text
Line 334: for how long
Response: for 4 days
Line 337: you need references
Response: reference was added
Line 347-357: how many replicates you used?
Response: three replicates were used that highlighted in text.
Line 366: this should be number
Response: this reference was corrected
Line 408: after how many days?
Response: 5 days that was mentioned at line 424 and highlighted.
Line 443:
Response: It means the times mentioned in the test, 0, 24, 72 and 96 h after inoculation of fungal inoculum.
Line 504: write reference here
Response: reference was added
Line 533: some references are old and need update and some need to fellow Journal style
Response: In our study we used some old references only in the introduction section to explain more about the topic and the review of the work history.
We look forward to hearing from you regarding our submission. We would be glad to respond to any further questions and comments that you may have.
Sincerely,
Jahanshir Amini
Round 2
Reviewer 1 Report
Thank you very much for your attention to my comments
Author Response
Dear Reviewer
We would like to express our gratitude to you for your valuable and constructive comments further improving our manuscript indeed.
With kind regards
Jahanshir Amini
Reviewer 2 Report
Authors covered all my comments
Author Response

(The authors gave the same response as above.)
